# 7,8-dimethoxycoumarin Attenuates the Expression of IL-6, IL-8, and CCL2/MCP-1 in TNF-α-Treated HaCaT Cells by Potentially Targeting the NF-κB and MAPK Pathways

**Nari Lee [1], You Chul Chung [1], Choon Il Kang [2], Sung-Min Park [3] and Chang-Gu Hyun [1,\*]**

[1]    Department of Chemistry and Cosmetics, Jeju National University, Jeju 63243, Korea
[2]    Jeju Indi Inc., Seogwipo-si, Jeju 63635, Korea
[3]    R&D Center, CoSeed Bio Pham Co., Chungbuk 28161, Korea
[\*]    Correspondence: cghyun@jejunu.ac.kr; Tel.: +82-64-754-3542

**Abstract:** 7,8-dimethoxycoumarin (DMC, C11H10O4), a natural coumarin compound, is present in *Citrus* plants including *Citrus decumana* and grapefruit. It is known to have protective effects on the kidneys against Cisplatin and ischemia-reperfusion injury. However, the underlying mechanisms of its inhibitory effects on skin inflammation have not been investigated in vitro. Tumor necrosis factor (TNF)-α is known to be one of the main causative agents of skin inflammation. It induces pro-inflammatory cytokines and chemokines by activating nuclear factor-κB (NF-κB) and mitogen-activated protein kinase (MAPK) signaling. In this study, we investigated the inhibitory effect of DMC on the expression of pro-inflammatory cytokines and chemokines in TNF-α-treated human keratinocyte HaCaT cells. Pretreatment with DMC inhibited TNF-α-treated cytokines (interleukin 6; IL-6) and chemokines (IL-8 and monocyte chemoattractant protein-1). In addition, DMC significantly inhibited TNF-α-treated NF-κB activation and phosphorylation of MAPKs, such as c-Jun N-terminal kinases (JNK) and extracellular-signal-regulated kinase (ERK). These results suggest that DMC may elicit an anti-inflammatory response by suppressing TNF-α-treated activation of NF-κB and MAPK pathways in keratinocytes. Hence, it might be a useful therapeutic drug against skin inflammatory diseases.

**Keywords:** 7,8-dimethoxycoumarin; human keratinocyte HaCaT cell; skin inflammation; NF-κB; MAPK

---

## 1. Introduction

The skin comprises three main layers, namely, epidermis, dermis, and subcutaneous fat layer. It is by far the largest organ in the body, protecting it from chemicals, disease, and ultraviolet and physical damage [1,2]. The epidermis, located surface-most on the skin, comprises five major layers: the stratum corneum, stratum lucidum, stratum granulosum, stratum spinosum, and stratum basale. It consists of a specific constellation of cell types, such as keratinocytes, langerhans cells, merkel cells, and melanocytes, of which keratinocytes are the predominant type, constituting approximately 95% of the cells. The keratinocytes can be found in the basal layer of the skin and are called basal cells or basal keratinocytes [3,4].

The primary function of epidermal keratinocytes is to provide a barrier against pathogens. Once the skin pathogens penetrate the epidermis, keratinocytes begin to produce various cytokines and chemokines such as TNF-α, IL-6, IL-8, and monocyte chemoattractant protein-1 (MCP-1), which lead to the pathogenesis of skin inflammation. These factors also increase infiltration of

immunocytes into the area of inflammation in the skin. Therefore, down-regulation of pro-inflammatory cytokine and chemokine production in keratinocytes can be an effective strategy in the treatment of inflammatory skin diseases [5–9]. According to various studies, transcription factors, such as nuclear factor kappa-light-chain-enhancer of activated B cells (NF-κB) and mitogen-activated protein kinase (MAPK), play critical roles in the cytokine and chemokine production mediated by TNF-α and/or interferon-gamma (IFN-γ) in human epidermal keratinocytes. Hence, these transcription factors act as both primary inducers and targets of the immune responses occurring in the keratinocytes. As NF-κB and MAPK are critical in skin immunopathology, the development of an effective strategy to block these pathways is a key step to controlling skin inflammation [10–12].

Currently, both steroids and non-steroidal anti-inflammatory drugs are prescribed clinically to reduce skin inflammatory conditions such as atopic dermatitis and psoriasis. They decrease chemokine and cytokine production by inhibiting the activity of NF-κB or MAPK pathways [13–16]. However, steroids are immunosuppressive agents that inhibit immunocyte functions, and long-term dosage of non-steroidal anti-inflammatory drugs can cause side effects, such as allergic reactions or drug resistance [4,12]. Therefore, the development of novel, complementary, and alternative drugs that can alleviate inflammatory skin diseases is of particular interest. Recent studies have found that flavonoids or coumarins isolated from plants may have potential therapeutic effects against skin inflammation [17–19].

7,8-dimethoxycoumarin (DMC, Figure 1a), a natural coumarin compound, is present in several medicinal plants, including *Daphne koreana, Astianthus viminalis, Zanthoxylum leprieurii,* and *Citrus decumana.* It is known to possess antioxidant and anti-inflammatory biological activities. DMC has been shown to protect the kidney against CP and I/R injury by virtue of these properties and inactivation of the mitochondrial permeability transition pore opening. DMC isolated from ethyl acetate extract of *C. decumana* has been shown to have an ameliorative effect on gastric inflammation [20–24].

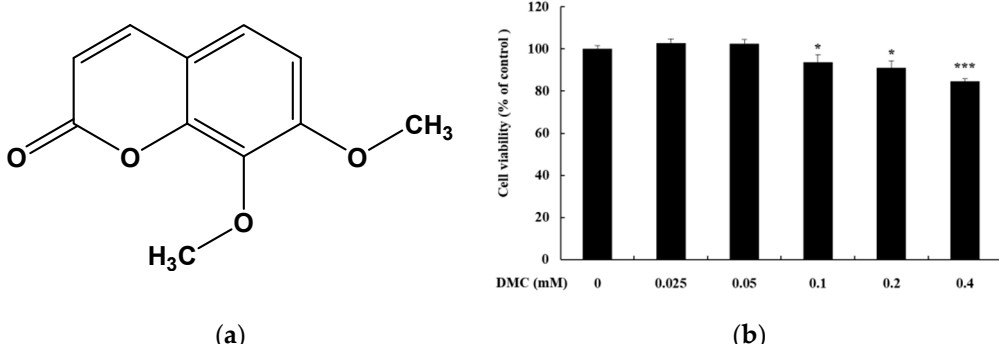

(**a**)                                                    (**b**)

**Figure 1.** Effect of 7,8-dimethoxycoumarin (DMC) on the viability of HaCaT cells. (**a**) Structures of 7,8-dimethoxycoumarin. (**b**) Cells were treated with DMC (0.025, 0.05, 0.1, 0.2, and 0.4 mM) for 20 h. Data are presented as mean ± standard deviation (SD) of at least four independent experiments (n = 4). * $p < 0.05$, *** $p < 0.001$ vs. control.

However, the anti-inflammatory effects and the basic mechanism of action of DMC in human keratinocytes have not yet been studied. Therefore, in this study, we examine the inhibitory effect of DMC on pro-inflammatory cytokine or chemokine production and the underlying molecular mechanism in TNF-α-treated human keratinocyte HaCaT cells. Our results suggest that DMC may exert anti-inflammatory effects by inhibiting the activation of NF-κB and MAPKs in keratinocytes, and hence the possibility of its application as a therapeutic agent for skin inflammation can be explored.

## 2. Materials and Methods

### 2.1. Chemicals and Reagents

7,8-dimethoxycoumarin (DMC, C11H10O4), tumor necrosis factor-$\alpha$ (TNF-$\alpha$), 3-(4,5-dimethylthiazol-2-yl)-2,5-diphenyltetrazolium bromide (MTT), ammonium pyrrolidinedithiocarbamate (APDTC), and protease inhibitor cocktail were purchased from Merck (Darmstadt, Hesse, Germany). Dulbecco's Modified Eagle medium, fetal bovine serum (FBS), penicillin-streptomycin (5000 U/mL), trypsin-ethylenediaminetetraacetic acid, BCA protein assay kit, and PD98059 (ERK inhibitor) were purchased from Thermo Fisher Scientific (Waltham, MA, USA). IL-6 IL-8, CCL2/MCP-1, and enzyme-linked immunosorbent assay (ELISA) kit were purchased from R&D Systems, Inc. (Minneapolis, MN, USA). Antibodies against P-p38(Thr180/Tyr182), T-p38, P-JNK(Thr183/Tyr185), T-JNK, P-ERK(Thr202/Tyr204), T-ERK, P-p105(Ser933), T-p105, P-p65(Ser536), T-p65, and $\beta$-actin were purchased from Cell Signaling Technology (Danvers, MA, USA). SP600125 (JNK inhibitor) and SB203580 (p38 inhibitor) were purchased from Cayman Chemical (Ann Arbor, MI, USA) and Calbiochem (San Diego, CA, USA), respectively. Dimethyl sulfoxide (DMSO), radioimmunoprecipitation (RIPA) buffer, enhanced chemiluminescence reagent (ECL kit), and 2×Laemmli sample buffer were obtained from Biosesang (Sungnam-si, Gyeonggi-do, Republic of Korea) and Bio-Rad (Hercules, CA, USA), respectively.

### 2.2. Cell Culture

The human keratinocyte cell line, HaCaT, was purchased from CLS Cell Lines Service GmbH (Eppelheim, Heidelberg, Baden-Württemberg, Germany). The cells were subcultured in Dulbecco's modified Eagle's medium containing 10% FBS and 1% penicillin-streptomycin at 2-day intervals and maintained in an incubator at 37 °C and 5% $CO_2$.

### 2.3. Cell Viability Assay

Cell viability upon DMC treatment was measured by MTT assay. HaCaT cells were seeded in a 24-well plate at $1.5 \times 10^5$ cells/mL and pre incubated for 24 h. The supernatant was removed, DMC (0.025, 0.05, 0.1, 0.2, and 0.4 mM) was added to the medium without FBS, and each concentration was cultured again for 24 h. Then, 400 μL of 0.4 mg/mL MTT solution was added to each well and incubated for 4 h. After completion of the reaction, the cell culture supernatant was removed; the formazan crystals formed were completely dissolved using DMSO, and the absorbance was measured at 570 nm using a microplate reader (SUNRISE, TECAN Austria GmbH).

### 2.4. Measurement of Cytokine Concentration

To investigate the effect of DMC on the production of cytokines such as IL-6, IL-8, and CCL2/MCP-1, HaCaT cells were seeded in a 24-well plate at $1.5 \times 10^5$ cells/mL and cultured for 24 h. Next, the supernatant was removed, DMC (0.025, 0.05, 0.1, and 0.2 mM) and TNF-$\alpha$ (50 ng/mL) were added along with FBS-free medium, and the cells were cultured for 24 h. Cell culture supernatants were collected, and the level of each of the cytokines was measured using an ELISA kit.

### 2.5. Western Blot Assay

HaCaT cells were seeded in a 60-mm culture dish at $3 \times 10^5$ cells/well and cultured for 24 h. The supernatant was removed, and various concentrations of DMC (0.05, 0.1, and 0.2 mM) and TNF-$\alpha$ (50 ng/mL) were added in the absence of FBS. The cultured cells were washed twice with 1×phosphate-buffered saline, lysed with lysis buffer (150 mM sodium chloride, 1% Triton X-100, 1% sodium deoxycholate, 0.1% SDS, 50 mM Tris-HCl pH 7.5, and 2 mM EDTA, sterile solution) and protease inhibitor cocktail (1.0%) at 4 °C for 20 min and centrifuged at 15,000 rpm for 20 min. Protein concentration was determined by quantitative analysis with the BCA protein assay kit, using bovine

serum albumin (BSA) as a standard. Proteins were quantitated, and 20 μg of each protein was loaded in an SDS-PAGE gel and then transferred to a polyvinylidene difluoride (PVDF) membrane. The membrane was blocked with 5% non-fat skim milk (blocking buffer) dissolved in tris-buffered saline containing 0.1% Tween 20 (TBS-T) for 2 h at room temperature and then washed thrice with TBS-T. Then, the primary antibody, dissolved at a ratio of 1:1000, was added to the blots and incubated for 24 h. After completion of the reaction, the blots were washed five times at 10 min intervals with TBS-T and then incubated with secondary antibody at a ratio of 1:3000 for 2 h. The blots were washed again with TBS-T five times at 10 min intervals. Proteins were detected using the ECL kit, and images were captured and analyzed using Chemidoc (Fusion solo 6S.WL, VILBER LOURMAT, France).

### 2.6. Statistical Analysis

The results of all experiments are expressed as mean ± standard deviation (mean ± SD). Statistical significance and *p*-values were calculated using the Student's *t*-test.

## 3. Results

### 3.1. Effect of DMC on Viability of HaCaT Cells

To confirm that the anti-inflammatory effect of DMC is not due to cell apoptosis, we first assessed the cytotoxicity of DMC on HaCaT cells using MTT assay. HaCaT cells were treated with various concentrations of DMC (0.025, 0.05, 0.1, 0.2, and 0.4 mM) for 20 h. The viability of the DMC-treated cells, compared to that of the untreated cells (control), was significantly different at 0.1, 0.2, and 0.4 mM concentrations. DMC did not show any toxic effects on cell viability up to 0.2 mM concentration (Figure 1b). In contrast, cell survival rate was 85% at a DMC concentration of 0.4 mM. Therefore, subsequent experiments were conducted at the highest concentration of DMC at 0.2 mM, which was not cytotoxic.

### 3.2. Effects of DMC on the Expression of Pro-Inflammatory Cytokines in TNF-α-treated HaCaT Cells

When keratinocytes are induced with TNF-α, which is known as a typical pro-inflammatory cytokine that causes inflammation, abnormal expression of cytokines and chemokines occurs. We, therefore, examined the inhibition effects of DMC on cytokine and chemokine production in TNF-α-treated HaCaT cells. The cells were pretreated with DMC (0.025, 0.05, 0.1, and 0.2 mM) for 1 h and then stimulated with TNF-α (50 ng/mL) for 20 h; subsequently, to determine the IL-6, IL-8, and CCL2/MCP-1 levels in cell supernatants, we used the ELISA kit. MTT assay was used to confirm the cell survival rate of DMC and TNF-α-treated controls (Figure 2a). TNF-α treatment significantly increased the production of IL-6, IL-8, and CCL2/MCP-1 in the cell supernatant, whereas pretreatment with DMC (0.025, 0.05, 0.1, and 0.2 mM) inhibited the production of cytokine or chemokine in a concentration-dependent manner (Figure 2b–d).

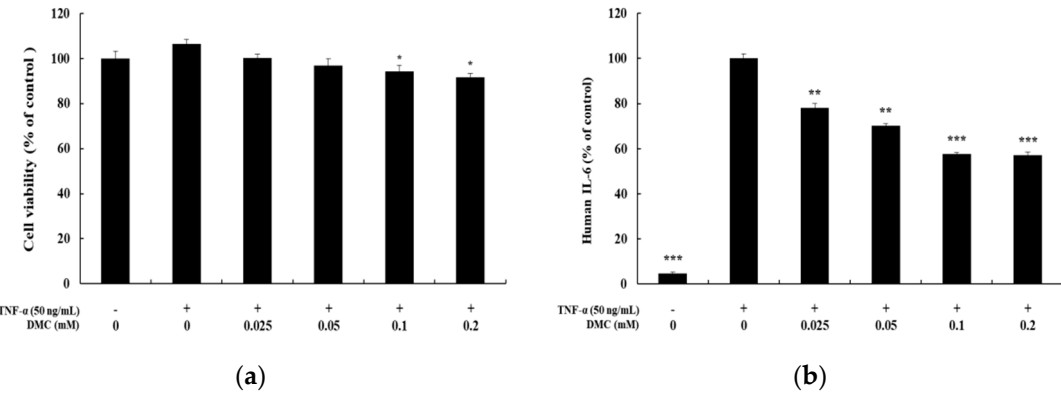

(a)　　　　　　　　　　　　　　　　　　　(b)

**Figure 2.** *Cont.*

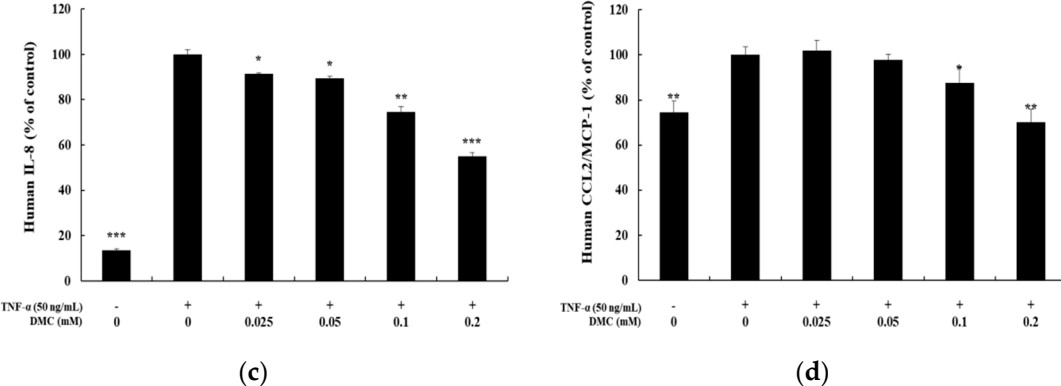

(c)　　　　　　　　　　　　　　　　　　　　(d)

**Figure 2.** Effects of DMC on the pro-inflammatory cytokine production and viability of tumor necrosis factor (TNF)-α-treated HaCaT cells. The cells were treated with DMC (0.025, 0.05, 0.1, and 0.2 mM) and TNF-α (50 ng/mL) for 20 h. TNF-α-treated cells were used as a positive control. (**a**) Cell viability was determined by 3-(4,5-dimethylthiazol-2-yl)-2,5-diphenyltetrazolium bromide (MTT) assay. (**b**); (**c**); and (**d**) production of IL-6, IL-8, and CCL2/MCP-1, were measured using the culture supernatant of HaCaT cells treated as mentioned above by ELISA. The results are expressed as percentage values compared to that of the control cells. Data are presented as mean ± SD (n = 4). * $p < 0.05$, ** $p < 0.01$, and *** $p < 0.001$ vs. non treated control or TNF-α- treated control.

### 3.3. Effect of TNF-α on the MAPKs and NF-κB Pathway in HaCaT Cells

In previous studies, it has been reported that HaCaT cells differ in the degree of phosphorylation of MAPKs and NF-κB signaling pathways upon TNF-α stimulation for different time periods [5,25,26]. Hence, before confirming the signaling pathway involved in mediating the anti-inflammatory effects of DMC, we examined the time at which maximum expression of phosphorylated MAPKs and NF-κB signaling intermediates is seen, by treating HaCaT cells with TNF-α (50 ng/mL) for 5, 10, 15, 30, and 60 min. The expression of P-p38, P-JNK, and P-ERK in the MAPK signaling pathway was the highest at 10 min after TNF-α treatment (Figure 3a–d). In addition, the expression of P-p105 and P-p65 in the NF-κB signaling pathway was also the highest at 10 min, and IκB-α was found to be degraded at 10 min (Figure 4a–d). Therefore, we treated cells with TNF-α for an optimum time of 10 min to determine the anti-inflammatory effects of DMC on MAPK and NF-κB signaling pathways.

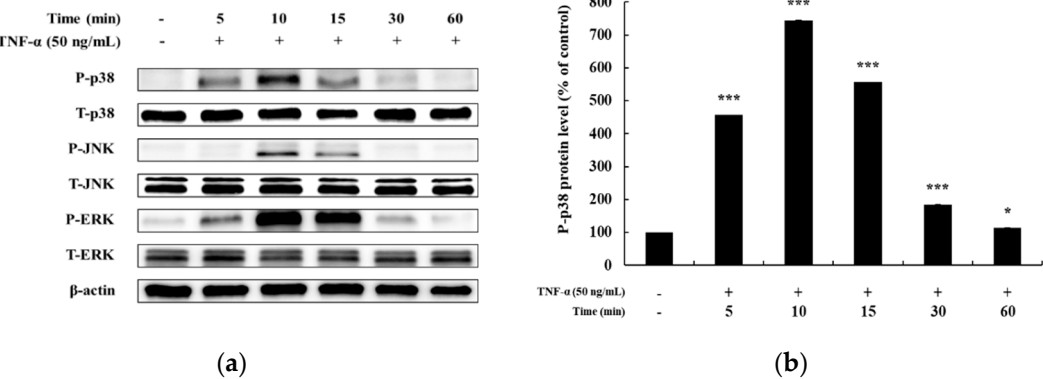

(a)　　　　　　　　　　　　　　　　　　　　(b)

**Figure 3.** *Cont.*

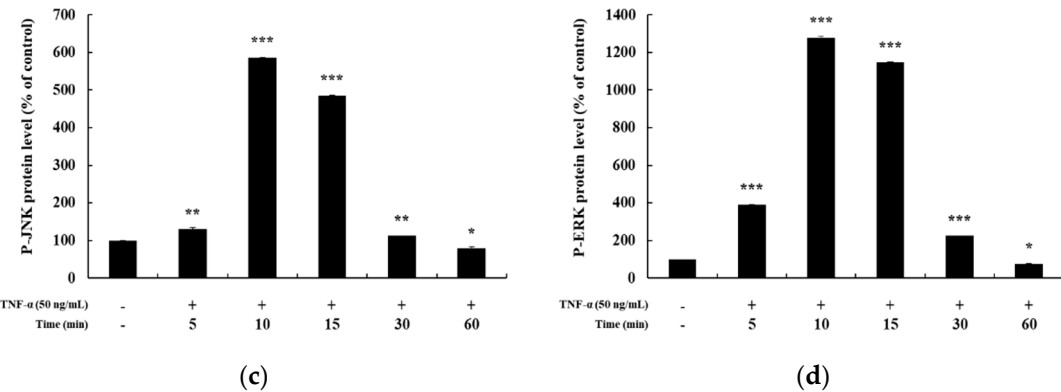

**Figure 3.** Effect of TNF-α on mitogen-activated protein kinase MAPK pathway in HaCaT cells. Cells were treated with TNF-α (50 ng/mL) for 5, 10, 15, 30, and 60 min, and cell lysates were prepared for western blot analysis. Data are presented as the mean ± SD (n = 3). * $p < 0.05$, ** $p < 0.01$, and *** $p < 0.001$ vs. control.

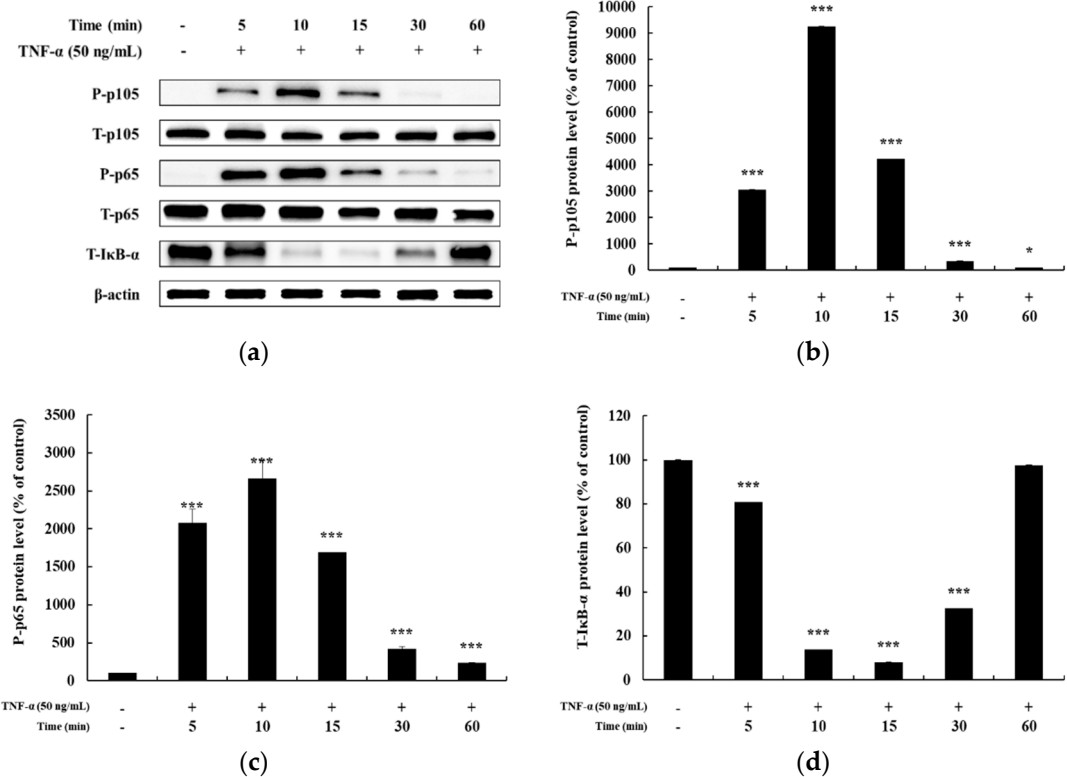

**Figure 4.** Effect of TNF-α on NF-κB pathway in HaCaT cells. Cells were treated with TNF-α (50 ng/mL) for 5, 10, 15, 30, and 60 min, and cell lysates were prepared for western blot analysis. Data are presented as the mean ± SD (n = 3). * $p < 0.05$, *** $p < 0.001$ vs. control.

### 3.4. Effect of DMC on the MAPK Pathway in TNF-α-Treated HaCaT Cells

Previously, TNF-α has been shown to activate the MAPK pathway in subsequent expression of pro-inflammatory mediators in keratinocytes [6]. To explore the correlation between MAPK pathway activation and pro-inflammatory mediators, we further investigated the regulatory effect of DMC on ERK, p38, and JNK phosphorylation by western blot analysis using phospho-specific antibodies. HaCaT cells were pretreated with SB203580 (p38 inhibitor, 10 μM), SP600125 (JNK inhibitor, 20 μM), and PD98059 (MEK1/2 inhibitor, 20 μM) for 1 h before TNF-α treatment. Compared to that in the control group, SB203580, SP600125, and PD98059 suppressed the TNF-α-treated phosphorylation of

MAPKs. We determined the alterations in the phosphorylation levels of MAPKs by calculating the P-ERK/T-ERK, P-p38/T-p38, and P-JNK/T-JNK ratios in TNF-α-treated HaCaT cells. DMC treatment reduced TNF-α-induced phosphorylation of ERK and JNK in a concentration-dependent manner without affecting the total protein levels, whereas it had a no effect on P-p38 levels. In particular, JNK and ERK phosphorylation were inhibited by 49% and 38%, respectively, at 0.1 mM DMC concentration compared to that in the TNF-α-treated control group (Figure 5). These results suggest that activation of p38, JNK, and ERK is partially involved in TNF-α-mediated stimulation of pro-inflammatory mediators, and that DMC can modulate MAPKs signaling cascades.

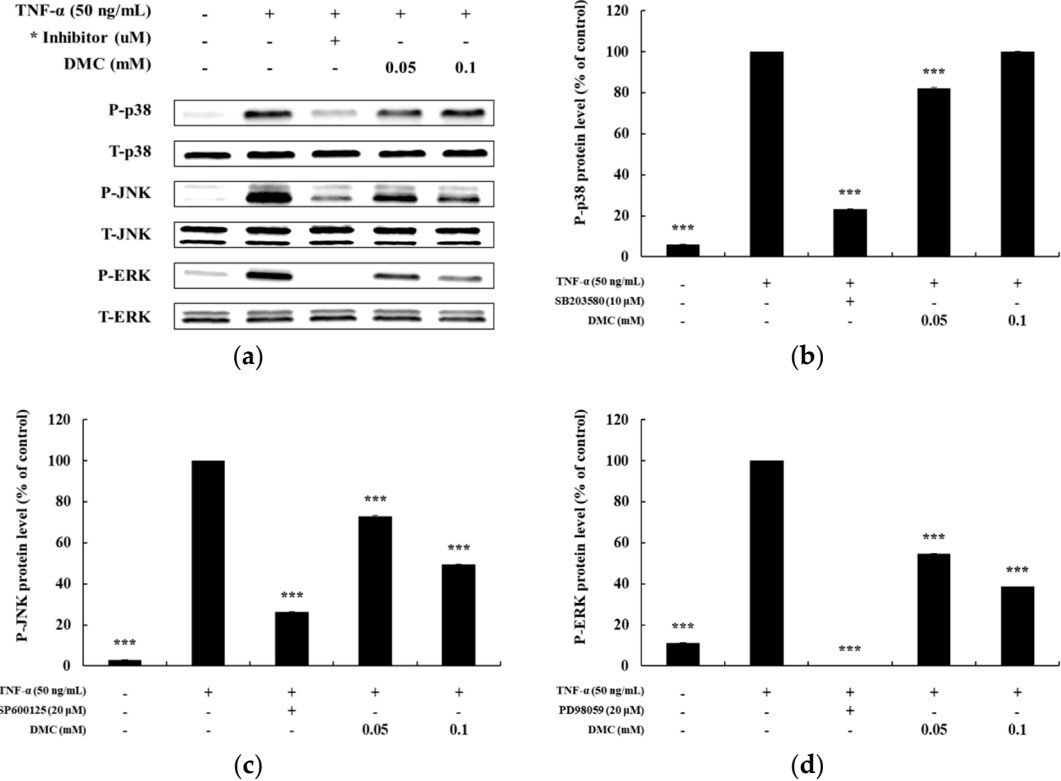

**Figure 5.** Effect of DMC on the expression of p38, c-Jun N-terminal kinases (JNK), and extracellular-signal-regulated kinase (ERK) in TNF-α-treated HaCaT cells. (**a**) cells were pretreated with various concentrations of DMC (0.05 and 0.1 mM) for 1 h and then stimulated with TNF-α for 10 min followed by western blotting of the cell lysates with the indicated antibodies. (**b**); (**c**); and (**d**) quantitation of the protein levels of P-p38 (b), P-JNK (c), and P-ERK (d), are presented as mean ± SD (n = 3). \*\*\* $p < 0.001$ vs. TNF-α- treated control.

### 3.5. Effect of DMC on the NF-κB Pathway in TNF-α Treated HaCaT Cells

Regulation of NF-κB signaling, like that of the MAPK pathway, is also an important therapeutic target for inflammation occurrence because activation of the NF-κB pathway relates to inflammatory disease. Previous studies have shown that IκB-α is degraded in the cytoplasm upon activation of the NF-κB signaling pathway, and that p65 and p105, which are associated with IκB-α, are activated in the cytoplasm by phosphorylation to promote inflammation [8]. Therefore, we examined whether DMC affected NF-κB pathway activation in TNF-α-treated HaCaT cells. As expected, P-p105 and P-p65 were expressed at high levels at 10 min in HaCaT cells stimulated with TNF-α. The treatment with the NF-κB inhibitor, APDTC (20 μM), also significantly decreased the levels of TNF-α-treated P-p105 and P-p65. Compared to the TNF-α-treated control group, treatment with DMC did not significantly affect TNF-α-treated P-p105 activation at concentrations of 0.05, 0.1, and 0.2 mM (Figure 6a). In contrast, it significantly inhibited the phosphorylation of NF-κB/p65 in a dose-dependent manner, compared to

that in the control group, indicating that DMC exerts its anti-inflammatory activity by inhibiting the signaling cascades leading to NF-κB activation (Figure 6b).

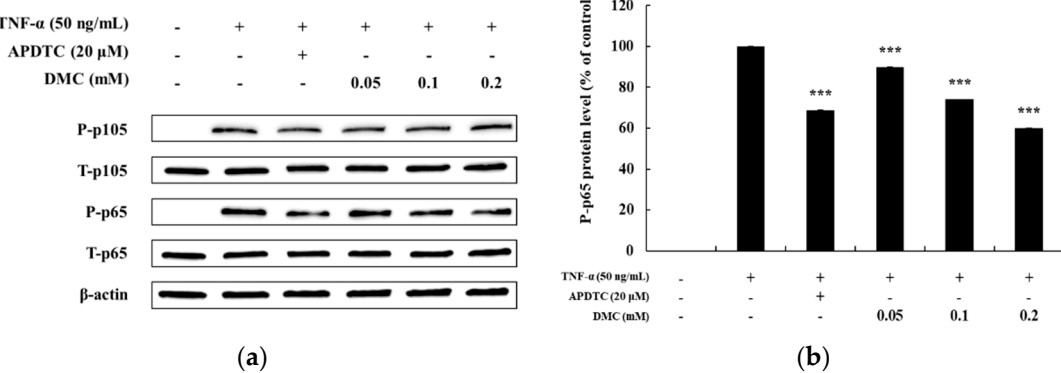

(**a**)　　　　　　　　　　　　　　　　　　　　　　　　　　(**b**)

**Figure 6.** Effect of DMC on p105and p65 expression in TNF-α-treated HaCaT cells. (**a**) Cells were pretreated with various concentrations of DMC (0.05, 0.1, and 0.2 mM) for 1 h and then stimulated with TNF-α for 10 min followed by western blotting of the cell lysates with the indicated antibodies. (**b**) Quantitation of protein level of P-p65 presented as mean ± SD (n = 3). *** $p < 0.001$ vs. TNF-α-treated control.

## 4. Discussion

Coumarin(1, 2-benzopyrone), comprising benzene and heterocyclic rings of 2-pyrone has been described as having anticoagulant, anticancer, antioxidant, antiviral, antidiabetic, anti-inflammatory, anti-bacterial, anti-fungal, and anti-neurodegenerative properties. The benzopyrones can be separated into benzo-alpha-pyrones, where coumarin belongs, and benzo-gamma-pyrones, of which flavonoids are the main component [27–29]. At present, they are also used in functional foods and cosmetic ingredients, as they possess anti-melanogenic and grey-hair preventing activities [30,31]. In addition, various coumarin compounds have been reported to exhibit anti-inflammatory effects [32–34].

The structural features, biosynthetic pathway, and biological activity of hundreds of natural coumarin compounds from plants have been studied. Thus, they also occupy an important place in the field of natural product and organic chemistry.

Many studies have shown that flavonoids with methoxy structures are more noticeable in functional and industrial aspects than flavonoids with hydroxy structures [35–37]. Therefore, we conducted research to determine the effect of DMC on skin inflammation. Previous studies have indicated that DMC exerts antioxidant and anti-inflammatory effects, thereby protecting the kidney against CP and I/R injury [20–24]. However, studies on the inhibitory activity of DMC against pro-inflammatory cytokines and chemokines have been limited. Therefore, we investigated the anti-inflammatory activity of DMC and the signaling pathways involved using the human keratinocyte HaCaT cell line.

Keratinocytes are resident cells that are mainly involved in the pathogenesis of allergic skin diseases and in the initiation and progression of allergic contact dermatitis. During sensitization, allergen induces activation of keratinocytes, causing the release of inflammatory mediators such as cytokines or chemokines. TNF-α, in turn, stimulates keratinocytes to produce and release various inflammatory proteins, including IL-8, IL-6, and MCP-1/CCL2 in an autocrine manner [5]. The unique, spontaneously immortalized, human keratinocyte HaCaT cells show properties similar to those of keratinocytes and are used in skin inflammatory symptoms and inflammatory in vitro tests for inflammatory stimuli such as TNF-α. Therefore, HaCaT cells can be an ideal system to confirm the biological activities of DMC in vitro [10,38].

IL-8 is a cytokine that belongs to the CXC chemokine family strongly stimulated by IL-1 and TNF-α. IL-8 has been found to be expressed in keratinocytes and is released in response to inflammatory stimulation. IL-6 is also a representative cytokine and is known to induce the proliferation and migration of keratinocytes to cause inflammatory skin diseases such as psoriasis and atopic dermatitis.

MCP-1 is a type of monocyte chemotactic activating factor, and it is a representative chemokine with regulated-upon-activation, normal T cell-expressed, and secreted (RANTES). MCP-1 is also known to be an important mediator of a variety of pathological conditions, including psoriasis and atopic dermatitis, and can be produced by fibroblasts, endothelial cells, mast cells, and keratinocytes [8,12,14]. Therefore, we studied the inflammation panel comprising the cytokine array, which indicated that DMC significantly inhibited the expression of chemokines and pro-inflammatory cytokines such as IL-8, IL-6, and MCP-1 in TNF-$\alpha$-stimulated HaCaT cells in a dose-dependent manner (Figure 2). These findings suggest that DMC may be useful to treat inflammatory skin diseases such as psoriasis and other related allergic diseases, based on its regulatory effects on chemokines and pro-inflammatory cytokines.

In keratinocytes, TNF-$\alpha$-stimulation activates MAPKs and NF-κB pathways, which play an important role in immune responses [5,8]. We investigated the inhibitory effect of DMC on the activation of the MAPK pathway in TNF-$\alpha$-treated keratinocytes. Our results showed that the phosphorylation of ERK and JNK is decreased as the concentration of DMC increases (Figure 5).

The production of pro-inflammatory cytokines and chemokines in keratinocytes could also be regulated by transcription factors in the NF-κB pathway. Previous studies have reported that NF-κB signaling pathways are activated by various stimulators such as TNF-$\alpha$. In addition, the activation of the NF-κB signaling pathway has been reported in psoriasis and atopic dermatitis [11,12]. Thus, the transcription factors of these NF-κB pathways play an important role in the synthesis of inflammatory cytokines and chemokines. Therefore, we performed experiments to evaluate the modulatory activity of DMC on the signaling pathways leading to NF-κB activation in TNF-$\alpha$-stimulated HaCaT cells. DMC significantly inhibited p65 phosphorylation in TNF-$\alpha$-stimulated HaCaT cells, indicating the modulation of the signaling cascades leading to NF-κB activation (Figure 6). In conclusion, the findings of the present study demonstrate for the first time that DMC inhibits the TNF-$\alpha$-treated expression of IL-6, IL-8, and CCL2/MCP-1 in HaCaT keratinocytes, and indicate that the inhibitory effect of DMC on the expression of these cytokines and chemokines is likely to be associated with the suppression of MAPKs (JNK and ERK) and NF-κB transcription factors. Our study suggests that DMC can be used as an alternative anti-inflammatory therapy for inflammatory skin diseases like psoriasis. Further studies on DMC therapy using in vivo models would be helpful in assessing the potential therapeutic effects on inflammatory skin.

**Author Contributions:** N.L. performed the experiments; Y.C.C., C.I.K., and S.-M.P. contributed reagents/materials/ analysis tools; C.-G.H. designed the experiments and wrote the paper.

**Acknowledgments:** This Study was supported by LINC$^+$ Development Project of Jeju National University (2018) through the National Research Foundation of Korea funded by the Ministry of Education

**Conflicts of Interest:** The authors declare no conflict of interest.

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
