# Peer review of "7,8-dimethoxycoumarin Attenuates the Expression of IL-6, IL-8, and CCL2/MCP-1 in TNF-α-Treated HaCaT Cells by Potentially Targeting the NF-κB and MAPK Pathways"

_cosmetics, doi:10.3390/cosmetics6030041_

Round 1

Reviewer 1 Report

In this manuscript, the authors described a attenuating activities of dimethoxycoumarin (DMC), contained in several medicinal plants and fruits, on TNF-alpha induced inflammatory-cytokine production such as IL-6, IL-8 and CCL2/MCP-1 in HACAT human keratinocytes. As mechanisms of action of DMC, authors reported DMC suppressed MAPKs and NF-kappaB pathway activated by TNF-alpha. They concluded DMC can be a candidate as a drug of anti-inflammatory threpy for inflammatory skin diseases. I think this manuscript contains some new findings. However, from the present data data, it seems that DMC does not have such strong activity. So I think some positive controls are necessary to compare the activities of DMC to other known compounds.

Author Response

Point 1:

However, from the present data data, it seems that DMC does not have such strong activity. So I think some positive controls are necessary to compare the activities of DMC to other known compounds

Response 1:  

Thank you for your useful comments and suggestions of our manuscript. We totally agree with editor’s opinion. In this environment, we would like to give two opinions.

First, in many recent similar articles, commercial chemicals are still not available for research. (Of course, we fully agree with your point of view). For reference, recent articles are listed below.

International Immunopharmacology Volume 72, July 2019, Pages 511-521

Cytokine. 2019 Jul;119:159-167. Epub 2019 Mar 23.

Journal of Ethnopharmacology 238,111829, 2019

Int Immunopharmacol. 2019 Jun;71:301-312. Epub 2019 Mar 29.

Int Immunopharmacol. 2019 Apr;69:270-278. Epub 2019 Feb 8.

Secondly, functional raw materials that are applied industrially are becoming an important indicator for industrialization of weight and price rather than molecular weight. Our research topic, DMC, is low molecular weight (MW 206.19) and low cost. For this reason, we believe that DMC is a valuable new functional material.

At present, we are totally dependent on your generous mind.

Reviewer 2 Report

The article is well structured, results are presented in an appropriate manner. 

Author Response

Point 1:

The article is well structured, results are presented in an appropriate manner.

Response 1:

We are thankful for your generosity.

Reviewer 3 Report

it is a good idea to tackle inflammatory responses associated with abnormal regulation of cutaneous biological functions using natural products and 7, 8-dimethoxycoumarin could be a potential additional armamentarium to the plethora of investigational agents.

However, several data put 

forward by the authors dampen enthusiasm for the modest outcome of this compound and the research results which if properly addressed shall make this work more vissible and accepted by the scientific community as follows.

Major:

the authors should revised the manuscript with the aid of professional english writers or native english user several areas in the manuscript are hard to follow and not clearly articulated.

The high concentrations of the drug that does not seem to show significant effect on cell viability as show in Fig 1., suggests complex mechanisms involved the authors should therefore show that apoptosis is involved at these concentrations or not and explain the significance of their data. these results should also be accompanied by a phase contrast image of cells showing morphology for better assessment.

In figure 2, authors should show the results of IL-1 (ALPHA AND BETA expression) in addition also include the effect of the compound treatment of HaCaT cells with stimulation. TPA could have been the best activator in this system as its a very well established agent for inducing KC initiation of proinflammatory signals.

the data in figure 3 does not seem to be the correct blots for some proteins analyzed why is pERK1/2 not a double band while the antibody from the vendor indicate it should be and the total erk1/2 also shows double band?.

Several result subtitles are not written in the right tenses as presented sometimes the verbs and adjectives are mixed though the content are correctly presented.

Does this agent have any effect of immune cell activation or proliferation?

Author Response

Point 1: the authors should revised the manuscript with the aid of professional english writers or native english user several areas in the manuscript are hard to follow and not clearly articulated.

Response 1: Thank you for your useful comments and suggestions on the structure of our manuscript. We totally agree with editor’s opinion. As you know, South Korea is not the English-speaking world. To avoid any grammar or syntax error, we have an additional plan for English editing of MDPI. Even though we received the English editing from Editage (www. Ediatge.com), we still have a lot of mistakes.

Point 2: The high concentrations of the drug that does not seem to show significant effect on cell viability as show in Fig 1., suggests complex mechanisms involved the authors should therefore show that apoptosis is involved at these concentrations or not and explain the significance of their data. these results should also be accompanied by a phase contrast image of cells showing morphology for better assessment. 

Response 2: We totally agree with editor’s opinion. In this environment, we would like to give two opinions. First, we searched the Pubmed system with the keyword "HaCaT & Inflammation". There were 653 papers available. On the other hand, with the keyword "HaCaT & Inflammation & apoptosis", 113 papers were found in the Pubmed system. Although our samples have been studied at high concentrations, there are many studies that do not include the results of apoptosis studies. Also, the 10-day fertilization period given to us is too much to cultivate HaCaT cells. Secondly, functional raw materials that are applied industrially are becoming an important indicator for weight and price rather than molecular weight. Our research topic, DMC, is low molecular weight (MW 206.19) and low cost. For this reason, we believe that DMC is a valuable new functional material. Finally, we are honored to publish our findings in the internationally renowned journal "Cosmetics" (SCOPUS). At present, we are totally dependent on your generous mind.

Point 3: In figure 2, authors should show the results of IL-1 (ALPHA AND BETA expression) in addition also include the effect of the compound treatment of HaCaT cells with stimulation. TPA could have been the best activator in this system as its a very well established agent for inducing KC initiation of proinflammatory signals. 

Response 3: As shown in the attached file, we have actually experimented with IL-1β several times. However, we did not observe an increase of IL-1β production in TNF-α-induced HaCaT cells. In this situation, we could not repeat all the experiments using TNF-α/IFN-γ together. At present, we are totally dependent on your generous mind.

Point 4: the data in figure 3 does not seem to be the correct blots for some proteins analyzed why is pERK1/2 not a double band while the antibody from the vendor indicate it should be and the total erk1/2 also shows double band?. 

Response 4: We can not make a clear interpretation why the band of total proteins seems to be two. What we can tell you is that bands of protein size expected in the Western experiment were clearly detected. In the case of phosphorylated proteins, faint bands are relatively invisible in the calibration of Western results.

Point 5: Several result subtitles are not written in the right tenses as presented sometimes the verbs and adjectives are mixed though the content are correctly presented. 

Response 5: As shown in the attached file, we received the English editing from Editage (www. Ediatge.com), To avoid any grammar or syntax error, we have an additional plan for English editing of MDPI.

Point 6: Does this agent have any effect of immune cell activation or proliferation? 

Response 6: We have found that DMC substances increase melanin production. We are currently submitting a paper to the MDPI Journal.

Round 2

Reviewer 1 Report

This revised manuscript has not improved what I pointed out.

Reviewer 3 Report

the authors have addressed most of my concerns

Author Response

We are thankful for your generosity.